# Wounds of Companion Animals as a Habitat of Antibiotic-Resistant Bacteria That Are Potentially Harmful to Humans—Phenotypic, Proteomic and Molecular Detection

**DOI:** 10.3390/ijms25063121

**Published:** 2024-03-08

**Authors:** Anna Lenart-Boroń, Klaudia Stankiewicz, Natalia Czernecka, Anna Ratajewicz, Klaudia Bulanda, Miłosz Heliasz, Daria Sosińska, Kinga Dworak, Dominika Ciesielska, Izabela Siemińska, Marek Tischner

**Affiliations:** 1Department of Microbiology and Biomonitoring, Faculty of Agriculture and Economics, University of Agriculture in Kraków, Adam Mickiewicz Ave. 24/28, 30-059 Krakow, Poland; klaudia.stankiewicz@student.urk.edu.pl; 2Scientific Circle of Biotechnologists, Faculty of Biotechnology and Horticulture, University of Agriculture in Kraków, 29 Listopada Ave. 54, 31-425 Krakow, Poland; natalia.czernecka@student.urk.edu.pl (N.C.); anna.ratajewicz@student.urk.edu.pl (A.R.); milosz.heliasz@student.urk.edu.pl (M.H.); daria.sosinska@student.urk.edu.pl (D.S.); 3Department of Forest Ecosystems Protection, Faculty of Forestry, University of Agriculture in Kraków, 29 Listopada Ave. 46, 31-425 Kraków, Poland; klaudia.bulanda@student.urk.edu.pl; 4Diagnostyka S.A. Medical Microbiological Laboratory, Na Skarpie 66, 31-913 Krakow, Poland; kinga.dworak@student.urk.edu.pl; 5The University Centre of Veterinary Medicine, Adam Mickiewicz Ave. 24/28, 30-059 Krakow, Poland; dominika.ciesielska@urk.edu.pl (D.C.); izabela.sieminska@urk.edu.pl (I.S.); 6Department of Animal Reproduction, Anatomy and Genomics, Faculty of Animal Science, Adam Mickiewicz Ave. 24/28, 30-059 Krakow, Poland; marek.tischner@urk.edu.pl

**Keywords:** antibiotic-resistant bacteria, antibiotic resistance genes, companion animals, veterinary medicine, wound infections

## Abstract

Skin wounds and their infections by antibiotic-resistant bacteria (ARB) are very common in small animals, posing the risk of acquiring ARB by pet owners or antibiotic resistance gene (ARG) transfer to the owners’ microbiota. The aim of this study was to identify the most common pathogens infecting wounds of companion animals, assess their antibiotic resistance, and determine the ARGs using culture-based, molecular, and proteomic methods. A total of 136 bacterial strains were isolated from wound swabs. Their species was identified using chromogenic media, followed by MALDI-TOF spectrometry. Antibiotic resistance was tested using disc diffusion, and twelve ARGs were detected using PCRs. The dominant species included *Staphylococcus pseudintermedius* (9.56%), *E. coli*, and *E. faecalis* (both n = 11, 8.09%). *Enterobacterales* were mostly resistant to amoxicillin/clavulanic acid (68.3% strains), all *Pseudomonas* were resistant to ceftazidime, piperacillin/tazobactam, imipenem, and tylosin, *Acinetobacter* were mostly resistant to tylosin (55.5%), all *Enterococcus* were resistant to imipenem, and 39.2% of *Staphylococci* were resistant to clindamycin. Among ARGs, *strA* (streptomycin resistance), *sul3* (sulfonamide resistance), and *blaTEM*, an extended-spectrum beta-lactamase determinant, were the most frequent. The risk of ARB and ARG transfer between animals and humans causes the need to search for new antimicrobial therapies in future veterinary medicine.

## 1. Introduction

In recent years, the number of pets in European countries has gradually increased, with dogs and cats becoming the two most prevalent types of companion animals [1]. The social role of companion animals has changed, and attention to their health and welfare has significantly increased, too [2]. Skin wounds are very common in small animals. Such wounds are frequently colonized by bacteria or show signs of bacterial infection. Both colonization and infection often result in healing delays and difficulties, thus increasing the cost of treatment [3]. Currently, the treatment of wounds in veterinary medicine is based on the administration of broad-spectrum antibiotics, the continuous administration of which contributes to the increasing antimicrobial resistance among both pathogenic bacteria and even commensal skin microbiota [4].

In this context, the threat of wound infection, especially by antimicrobial-resistant bacteria (ARB), becomes an even more serious problem. This is due to the risk of pet owners acquiring the ARB or antibiotic resistance genes (ARGs) via contact with their companion animals [2]. The risk of interspecies transmission of bacteria, including antibiotic-resistant or multidrug-resistant bacteria, is very high because the majority of bacterial pathogens in companion animals are bacterial species that can also commonly occur in humans. These pathogens and opportunistic pathogens are *Staphylococcus* (including *S. aureus*), *Pseudomonas* (including *P. aeruginosa*), beta-haemolytic *Streptococcus*, *E. coli*, or *Enterococcus* (including *E. faecalis* or *E. faecium*) [1,2,3,4,5,6].

With respect to the high risk of companion animals’ infections with pathogenic ARB, the risk of infection treatment failure is also high. There are surprisingly few studies concerning the scope of antimicrobial resistance in companion animals, while research on animal wound infections is limited to only a few texts [3,4]. For this reason, understanding the prevalence of companion animals’ wound infections caused by antimicrobial-resistant bacteria that can easily be transmitted to their owners or veterinarians is among the most important gaps to be filled. With this in mind, the aim of this study was to identify the most common pathogens infecting wounds of companion animals treated in veterinary clinics, assess their resistance to antibacterial agents used in their treatment, and determine the genes determining the most common and dangerous resistance mechanisms. The combination of culture-based, molecular biology, and proteomic methods was used in order to obtain the most reliable results.

## 2. Results

A total of 136 bacterial isolates were obtained from the collected swabs, including 80 (58.82%) from dogs, 52 (38.24%) from cats, and 4 (2.94%) from rabbits. There were 71 (52.21%) Gram-positive and 65 (47.79%) Gram-negative isolates. The dominant genera (Table 1) were *Staphylococcus* spp. (n = 37, 27.21%), followed by *Enterococcus* spp. (n = 17, 12.50%), *Escherichia* (n = 11, 8.09%), and *Acinetobacter* (n = 10, 7.35%), with the following dominant species: *S. pseudintermedius* (n = 13, 9.56%), and *E. coli* and *E. faecalis* (both n = 11, 8.09%). The dominant bacterial genera and species differed between the examined groups of animals (Table 1 and Appendix A, respectively).

Antimicrobial resistance of 120 bacterial isolates was tested using 16 antibacterial agents in five combinations based on the target organism (Table 1 and table in Section 4.3). Among the group of *Enterobacterales*, resistance to amoxicillin/clavulanic acid was the most prevalent (detected in 68.3% of strains), followed by ampicillin (52.6%) and tylosin (44.7%). The extended-spectrum beta-lactamase (ESBL) phenotype of resistance was detected in 13.16% of strains (Appendix A and Figure 1A). None of the examined antimicrobials were effective against all strains. All strains of *Pseudomonas* spp. (n = 7) were resistant to ceftazidime, piperacillin/tazobactam, imipenem, and tylosin. Only amikacin proved effective against all strains of *Pseudomonas* spp. (Figure 1B). Finally, *Acinetobacter* (n = 9) showed the highest percentage of resistance to tylosin (55.5% of strains), followed by enrofloxacin and gentamicin (both 44.4%, Figure 1C). In the case of Gram-positive bacteria, all *Enterococcus* spp. strains (n = 15) were resistant to imipenem, 66.7% were resistant to tigecycline, and 46.7% to enrofloxacin (Figure 2A). In *Staphylococcus* spp., resistance to clindamycin was the most frequently observed (i.e., in 39.2% of strains). Methicillin resistance (MRS), shown by resistance to cefoxitin, was the second most prevalent, and was observed in 33.3% of strains (Figure 2B), along with resistance to erythromycin (also detected in 33.3% of strains). Both macrolide/streptogramin b (MSb) and macrolide/lincosamid/streptogramin b (MLSb) constitutive phenotypes of resistance were observed in 13.7%, while the inducible MLSb phenotype was observed in 3.9% of strains (Appendix A; Figure 2B).

One *Enterobacterales* strain (canine *Proteus mirabilis*), out of all examined, was resistant (or insusceptible) to eight out of nine antimicrobial agents tested. Two other Gram-negative strains (*K. pneumoniae* and *Aeromonas media*) were resistant to seven antimicrobials (Table 2). Among Gram-positives, three *Staphylococci* were resistant to seven out of eight antimicrobials tested (Table 2; these were feline *S. pseudintermedius* and *S. aureus*, and canine *S. pseudintermedius*).

Finally, PCR tests were carried out to search for 12 genetic determinants of antibiotic resistance to all antimicrobial classes used in the treatment of Gram-positive and Gram-negative bacteria. Due to the fact that various resistance mechanisms may be characteristic of Gram-positive and Gram-negative bacteria, some genes examined in these two groups of bacteria varied (such as methicillin and MLSb resistance determinants assessed only in Gram-positives and ESBL determinants assessed only in Gram-negatives), while others (such as *strA* determining the aminoglycoside resistance and *sul3* that determines the resistance to sulfonamides) were common for both of these groups.

Ten out of the tested twelve genetic determinants of antimicrobial resistance were detected in the DNA extracted from bacterial isolates from cats, dogs, and rabbits (Table 3 and Table 4). Figure 3 presents the selected, most representative pictures showing the PCR results for the most frequently detected genes. Genes *ereA* (erythromycin esterase), *qnrA* and *qnrD* (plasmid-mediated quinolone resistance genes) were not detected in any of the examined samples. On the other hand, the streptomycin resistance gene, *strA* (Figure 3A), was detected in 29 samples, in all groups thereof, in both Gram-positive and Gram-negative bacteria. In terms of the detection frequency, it was followed by *sul3* (Figure 3B), a sulfonamide resistance gene (14 positive samples in Gram-positive and Gram-negative isolates of both cats and dogs), and *blaTEM* (Figure 3C), an ESBL determinant characteristic of Gram-negative bacteria (13 positive samples in cats and dogs, Table 4 and Figure 3A).

Interestingly, there have been six canine and one feline bacterial isolates with phenotypic and molecular resistance profiles (Table 5) that could make them alarming, as they were resistant to all or nearly all examined antimicrobial agents and their DNA contained three or four different ARGs.

## 3. Discussion

There is growing evidence that resistant bacteria (including multidrug-resistant) occur in companion animals and that many species among them are shared between animals and humans [2]. With this in mind, the inappropriate use of antimicrobials in animals may result in the selection and spread of antimicrobial resistance, thus constituting a potential risk to public health [2]. This study provides data on bacteria colonizing wounds of companion animals, their antimicrobial resistance profiles, and genetic determinants of resistance to all classes of antimicrobial agents among Gram-negative and Gram-positive bacteria. The distribution of bacterial taxa shows that *Staphylococcus* spp. (with *S. pseudintermedius* and *S. aureus*), *Enterococcus* spp. (with *E. faecalis*), *E. coli*, *Acinetobacter* spp. (with *A. ursingii*), and *Pseudomonas* (with *P. aeruginosa*) were the most frequently isolated from all samples. A high prevalence of *Staphylococcus* spp., with higher numbers of *S. pseudintermedius* than of *S. aureus,* was reported by [6,7]. Kožár et al. [3] also observed that *Staphylococcus* (including *S. intermedius*, later reclassified to *S. pseudintermedius*) was the most frequent among wound-infecting Gram-positive bacteria, while *E. coli* was the most frequently identified Gram-negative bacterium. *Staphylococcus pseudintermedius* is the most common canine bacterial pathogen, but it is, indeed, accompanied by a variety of other bacteria—both Gram-positive and Gram-negative bacteria.

In recent years, increased attention has been paid to the welfare of small companion animals, which results in, e.g., increased expenses on veterinary care. This involves the frequent use of antimicrobial agents in pets, and many of these antimicrobials are commonly used in both human and veterinary medicine. However, the identification of causal agents and their antimicrobial susceptibility is often neglected, leading to inappropriate empirical treatment [8]. In both cats and dogs, the most frequent causes of antimicrobial use are wound infections [8]. The most frequently used classes of antibiotics in animals are fluoroquinolones, β-lactams, cephalosporins, sulfonamides, macrolides, and glycopeptides [9]. Among the consequences of antimicrobial use in companion animals is that the amounts and patterns of antibiotic administration are reflected in the rate at which resistance develops and spreads in the exposed bacterial population [8].

In our study, a high percentage of *Enterobacterales* was resistant to β-lactam antibiotics (amoxicillin/clavulanic acid and ampicillin—68.3% and 52.6%, respectively), while 100% of *Pseudomonas* isolates were insusceptible to ceftazidime (cephalosporin), piperacillin/tazobactam (β-lactam), imipenem (carbapanem), and tylosin (macrolide). However, resistance to enrofloxacin (fluoroquinolone) ranged from 11.8% in *Staphylococci* to 46.7% in *Enterococci,* and it was never the highest—only in *Acinetobacter* spp., it was the third highest (44.4%) among the observed resistance rates. Generally, enrofloxacin is commonly used systemically in the infection treatment of small animals, and although it is still efficient, there have been cases of treatment failure [8]. With respect to the fact that inappropriate use (e.g., pulse-dose, low-dose) is very common in the treatment of small animals, this might favor the development of resistant strains, particularly when long-term treatment is required [10]. As far as the antibiotic administration effect is concerned, it has been reported that enrofloxacin treatment promotes multidrug-resistant (MDR) *E. coli* colonization and that the proportion of dogs carrying resistant *E. coli* increased with the duration of hospital stay and with the antimicrobial treatment [2].

Even though antimicrobial resistance is spread among environmental microorganisms and human and animal pathogens, resistance against last-resort antimicrobial agents for human medicine detected in microorganisms that can be easily transmitted between animals and humans seems of special concern [2,8]. Insusceptibility to imipenem (carbapenem antimicrobial) has been detected quite frequently in our study. All *Enterococcus* spp. and all *Pseudomonas* spp. isolates were resistant to this antimicrobial. Importantly, both canine *E. faecalis* MDR strains, positive for four genetic determinants of resistance and resistant to four antibiotics mentioned in Table 5, were also resistant to imipenem. In the case of Gram-negative isolates mentioned in the special concern group (Table 5), one canine *E. coli* and *P. mirabilis* were also imipenem-resistant. All these species are listed as human pathogens, too. They cause, among many others, urinary tract infections [8,11]. Close interaction between pets and humans favors the transmission of bacteria by both direct contact and through the domestic environment, which puts children at greater risk than adults. This is due to their more common and closer physical contact with pets and with the household environment [8]. The most important hazard for human health in the case of dealing with animals’ wounds is therefore related to the transmission of antimicrobial-resistant bacteria from pets to humans and the related zoonotic infections [2].

Notably, antimicrobial resistance can be transmitted by low bacterial numbers or even in the absence thereof, if only the genetic determinants are present in the environment. Resistance gene transfer frequently occurs horizontally. Most classes of antibiotics have long been used in both human and veterinary medicine; thus, the same resistance genes are being identified in bacteria isolated from humans and animals [8]. In our study, ten genetic determinants of antimicrobial resistance have been identified in bacterial isolates of cats, dogs, and rabbits. Streptomycin resistance (*strA*) was most frequently identified, followed by sulfonamide resistance (*sul3*) and one of the ESBL determinants (*blaTEM*). Studies on the genetic determinants in bacteria isolated from companion animals are scarce. However, the listed genes have also been mentioned by other authors to occur in bacteria isolated from companion animals [8,12,13]. All four examined ESBL determinants were detected in this study at the following frequency: *blaTEM* > *blaSHV* > *blaCTX m* > *blaOXA*. Carvalho et al. [12], on the other hand, detected *blaTEM*, *blaCTX m*, and *blaSHV* in similar proportions in *E. coli* isolates obtained from dogs and their owners. Akhtardanesh et al. [13] examined the presence of nine genes determining the resistance to tetracyclines, quinolones, aminoglycosides, sulfonamides, and trimethoprim in *E. coli* isolates from pet cats. Similarly, as in our study, *qnr* genes (determining quinolone resistance) were very rare or absent. In this study, resistance to enrofloxacin, which is the most frequently applied fluoroquinolone veterinary antibiotic, varied and ranged from 11.7% in staphylococci to 46.7% in enterococci. In an earlier study focused on antimicrobial resistance and the molecular resistance mechanisms in chicken feces-derived *E. coli*, Lenart-Boroń et al. [14] observed that the fluoroquinolone resistance would reach even 93.3% of isolates. At the same time, resistance to β-lactams was also very frequent, with some experimental groups reaching 100% of bacterial isolates. Consequently, *qnrB* and *qnrS* genes were very frequent (detected in more than 81% and 86% of isolates of some experimental groups, respectively). However, the selection pressure put on bacteria present in farm poultry is much higher than on bacteria derived from pet animals. The reason for such a situation is that, despite the restrictions resulting from European Union regulations, the doses of antimicrobial agents administered to poultry are often inflated, used contrary to the veterinarians’ instructions, or the treatment duration is extended without justification [14].

## 4. Materials and Methods

### 4.1. Collection of Samples

A total of 145 wound swabs were collected from companion animals (84 dogs, 60 cats, and 1 rabbit) that underwent treatment at the University of Agriculture Veterinary Clinic. The samples were collected with a sterile swab by a qualified veterinarian (Appendix A), immediately transferred to the Laboratory of Microbiology and Biomonitoring, and inoculated on selective media for the isolation and preliminary identification of bacterial pathogens and opportunistic pathogens.

### 4.2. Isolation and Identification of Bacteria

The culture media included: Chromogenic UTI Medium (ThermoFisher Scientific, Oxford, UK) for presumptive identification and differentiation of *Enterococcus*, *Escherichia coli*, *Proteus*, *Pseudomonas*, *Staphylococcus*, and *Klebsiella*; MacConkey agar (Biomaxima, Lublin, Poland) for Gram-negative *Enterobacterales*; Tryptone-bile-X-glucuronide agar (Biomaxima, Lublin, Poland) for *E. coli*; Slanetz-Bartley agar (Biomaxima, Lublin, Poland) for Enterococcus; Baird Parker agar (Biomaxima, Lublin, Poland) for *Staphylococcus aureus*; Columbia CNA agar with 5% sheep blood (ThermoFisher Scientific, Oxford, UK) for Gram-positive cocci and determination of their haemolysis type; Cetrimide agar (Biomaxima, Lublin, Poland) for *Pseudomonas*. All cultures were conducted for 24–48 h at 37 ± 1 °C. After incubation, different morphotypes, characteristic of the examined groups of bacteria (Figure 4), were subcultured by plate streaking, followed by observations of Gram-stained preparations. The species identification was performed via MALDI-TOF (Matrix-assisted laser desorption/ionisation time-of-flight) technology on a Bruker microflex ^®^ mass spectrometry instrument (Bruker, Billerica, MA, US). This technique uses a laser to ionize and separate molecules by their mass-to-charge ratio. The resulting peptide mass fingerprint can be compared to a database of known spectra to find the best match. The high throughput, accuracy, and small required sample size of the MALDI-TOF MS have made it a widely used technique in the clinical microbiological laboratory. All identified bacteria had a log (score) value higher or equal to 2.0, which indicated ‘highly probable species identification’.

### 4.3. Antibacterial Susceptibility Tests

A total of 136 bacterial strains were isolated from the examined swabs. The antimicrobial susceptibility patterns were assessed using the disc-diffusion method, based on the recommendations of the Polish National Reference Centre for Antimicrobial Susceptibility [15]. Some species (e.g., *Brevundimonas diminuta*, *Hafnia alvei*, *Bacillus pumilus,* or *Curtobacterium flaccumfaciens*, Appendix A) proved either commensal or typically environmental. For this reason, international and Polish recommendations for antimicrobial susceptibility testing are not available, and neither is the interpretation of results. Therefore, the examinations were carried out on 120 isolates of Gram-negative (n = 54) and Gram-positive (n = 66) bacterial pathogens and potential pathogens. As the examined bacteria were classified into 28 different genera, the recommendations for both Gram-positive and Gram-negative bacteria were applied and divided into five groups (Table 6). Antimicrobial disc cartridges were obtained from Oxoid (Basingstoke, Great Britain). The ESBL (extended-spectrum beta-lactamase)-positive *Enterobacterales* and *Pseudomonas* strains were confirmed with the double disc synergy test [16]. The resistance of macrolide, lincosamid, and b-type streptogramin was assessed according to [17].

After incubation for 18–24 h at 36 ± 1 °C, the growth inhibition zone diameters around discs were measured (in mm) and compared to the most recent breakpoint values provided by the European Committee on Antimicrobial Susceptibility Testing [18].

### 4.4. Assessment of Genes Conferring the Bacterial Resistance to Different Groups of Antimicrobials

Bacterial DNA was extracted from all 120 strains subjected to antimicrobial resistance testing and from the control strains: susceptible *E. coli* ATCC 25922, Methicillin-susceptible *S. aureus* ATCC 25923; environmental strains of methicillin-resistant *S. aureus* and ESBL-positive *E. coli* were used as positive controls. DNA extraction was conducted from overnight cultures, using the Genomic Mini DNA extraction set of reagents (A&A Biotechnology, Gdańsk, Poland), following the manufacturer’s recommendations.

PCR amplifications were conducted using specific primers (Table 7) to search for genes conferring methicillin resistance in staphylococci, antibiotics from groups of macrolides, lincosamids, and streptogramins in Gram-positive cocci, ESBL determinants in *Enterobacteriaceae* and *Pseudomonas*, aminoglycosides, carbapenems, fluoroquinolones, sulfonamides, and tetracyclines in both Gram-negative and Gram-positive isolates. The reactions were performed in a volume of 25 μL containing 50 ng of DNA template, 12.5 pM of each primer, and 2× (12.5 μL) of PCR Mix Plus Green (A&A Biotechnology, Gdańsk, Poland) filled up with ultrapure water up to 25 μL. The following temperature profile was used for the reactions: initial denaturation at 95 °C for 5 min, followed by 35 cycles of 94 °C for 45 s, annealing for 45 s at temperatures corresponding to individual primers, then extension at 72 °C for 1 min, with the final extension at 72 °C for 10 min. The reactions were performed in a T100 Thermal Cycler (Bio-Rad, Hercules, CA, USA). The PCR products were electrophoresed for 60 min in 1 × TBE SimplySafe (EurX, Gdańsk, Poland)-stained 1% agarose gel and visualized in UV light. DNA size Marker 3 (A&A Biotechnology, Gdańsk, Poland) was used to assess the size of product bands.

### 4.5. Statistical Analysis

The significance of differences in bacterial resistance to various antimicrobial agents and the presence of genetic determinants of antimicrobial resistance were assessed using the chi-square test (https://www.socscistatistics.com/tests/chisquare2/default2.aspx, accessed 3 December 2023). The significance level was set at a *p* value of < 0.05 for all tests.

## 5. Conclusions

The issues of infection control, antimicrobial resistance, and the spread of resistance genes are the same in companion animal and human hospitals, as these facilities are characterized by intensive use of antibiotics and a high density of patients. For these reasons, they are high-risk environments for the occurrence and spread of nosocomial infections, resistant bacteria, and genetic determinants thereof [2,29]. As shown in the current study, the composition of bacterial species that infect companion animals’ wounds can vary greatly, and even though some species could be considered more prevalent than others, none of them were prevalent enough to conclude that their presence in pets’ wounds could be assumed with high probability. Moreover, these bacteria can carry a number of resistance genes that determine their insusceptibility to a variety of antimicrobial agents. This study also resulted in two disturbing observations: the first refers to the prevalence of multidrug-resistant bacteria (in some groups reaching even 100% of isolates), while the second is the detection of bacteria that are also human pathogens (e.g., *Proteus mirabilis*), which were resistant to nearly all or all antimicrobial agents tested and carried three or four antibiotic resistance genes. The widely reported risk of transfer of resistant bacteria between animals and humans, coupled with the increasing demand for advanced therapies in companion animals and the spread of MDR bacteria, may result in the need to search for new antimicrobial therapies in the future in veterinary medicine.

## Figures and Tables

**Figure 1 ijms-25-03121-f001:**
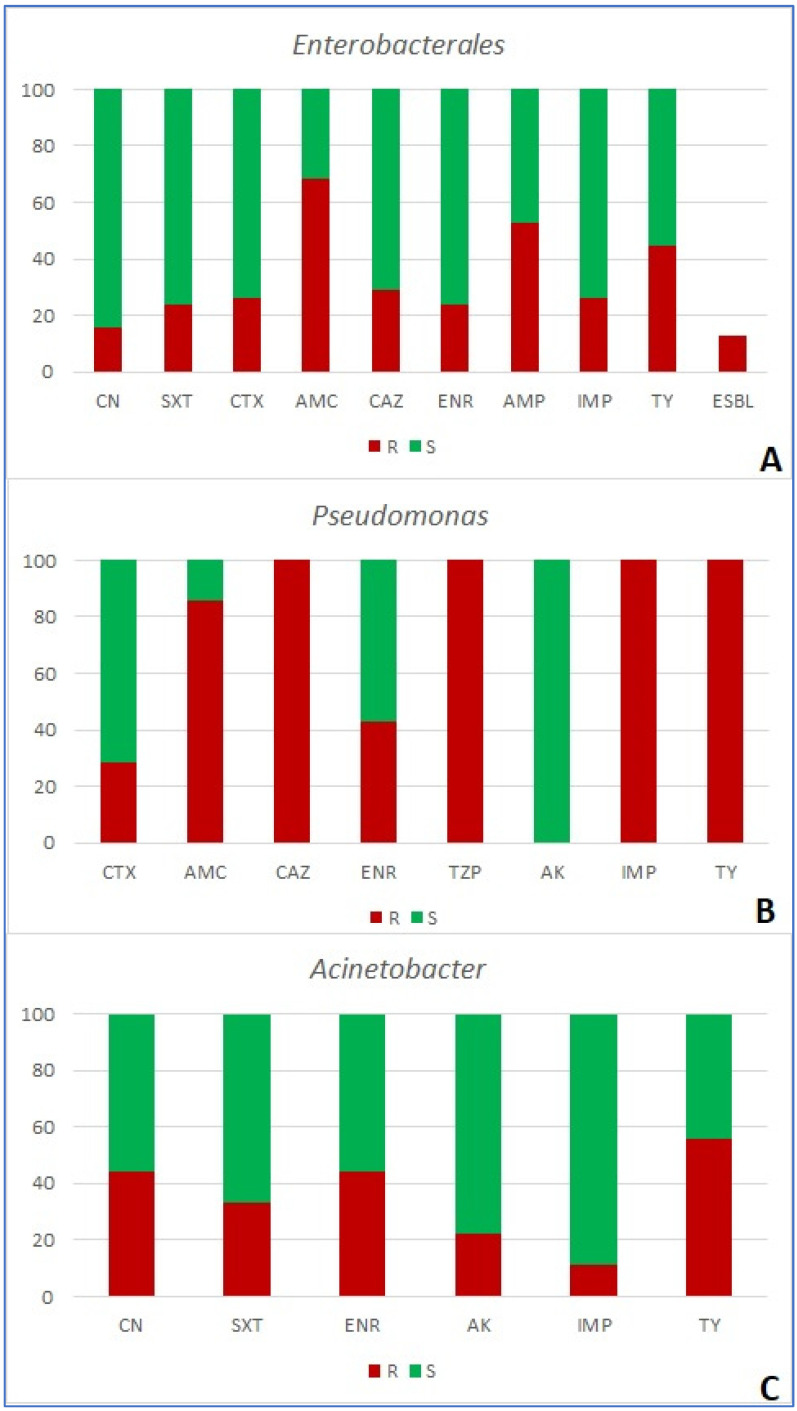
Share (%) of resistant (R) and susceptible (S) Gram-negative bacterial strains ((**A**)—*Enterobacterales*; (**B**)—*Pseudomonas*; (**C**)—*Acinetobacter*) isolated from wounds of companion animals. CN—gentamycin; SXT—trimethoprim/sulfamethoxazole; CTX—cefotaxime; AK—amikacin; AMC—amoxicillin/clavulanic acid; CAZ—ceftazidime; ENR—enrofloxacin; AMP—ampicillin; IMP—imipenem; TY—tylosin; TZP—piperacillin/tazobactam; ESBL—extended-spectrum-beta-lactamase-producing strains of *Enterobacterales*.

**Figure 2 ijms-25-03121-f002:**
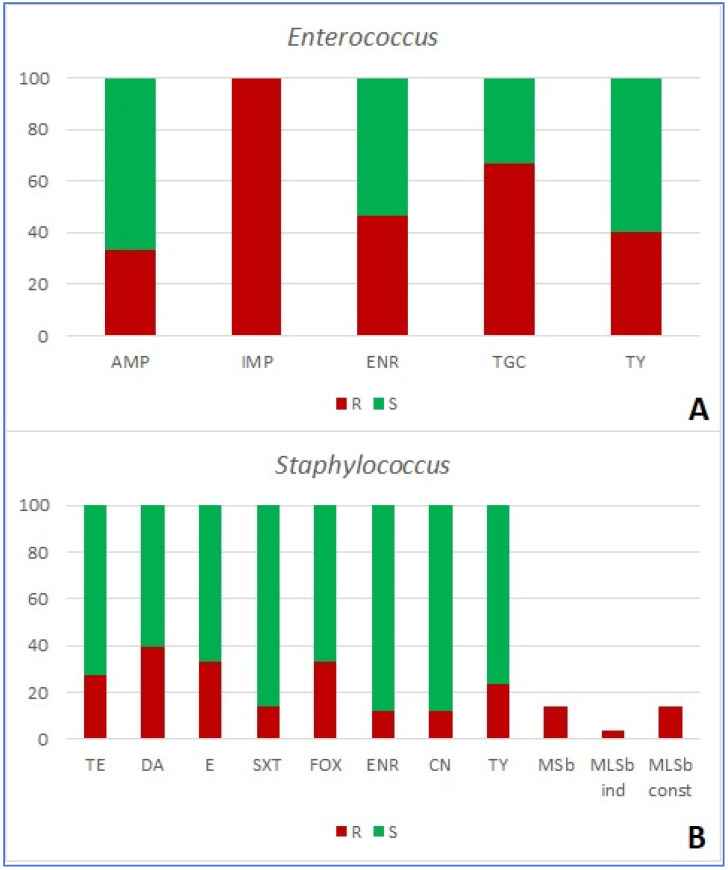
Share (%) of resistant (R) and susceptible (S) Gram-positive bacterial strains ((**A**)—*Enterococcus*; (**B**)—*Staphylococcus*) isolated from wounds of companion animals. AMP—ampicillin; CN—gentamycin; DA—clindamycin; E—erythromycin; IMP—imipenem; ENR—enrofloxacin; FOX—cefoxitin; SXT—trimethoprim/sulfamethoxazole; TGC—tigecycline; TE—tetracycline; TY—tylosin; MSb—resistance mechanisms to macrolides and streptogramins b; MLSb ind—inducible mechanisms of resistance to macrolides, lincosamids, and streptogramins b; MLSb const—constitutive mechanisms of resistance to macrolides, lincosamids, and streptogramins b.

**Figure 3 ijms-25-03121-f003:**
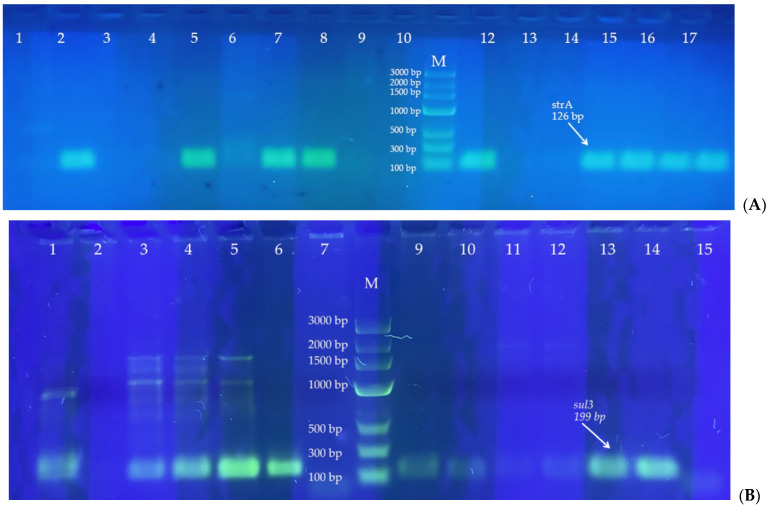
Positive results of the PCR test for the exemplary bacterial isolates, the three most frequently detected genes in Gram-positive and Gram-negative bacteria. Lanes marked as M—DNA Marker 3 (A&A Biotechnology, Poland, Gdańsk); lanes 1–17 (**A**), 1–15 (**B**) and 1–18 (**C**)—bacterial DNA isolates; lanes 2, 5, 7, 8, 12, 14–17—DNA bands of searched length, confirming the presence of strA gene (**A**); lanes 1, 3–6, 9–14—DNA bands of searched length, confirming the presence of sul3 gene (**B**); lanes 5–7 and 13–18—DNA bands of searched length, confirming the presence of *blaTEM* gene.

**Figure 4 ijms-25-03121-f004:**
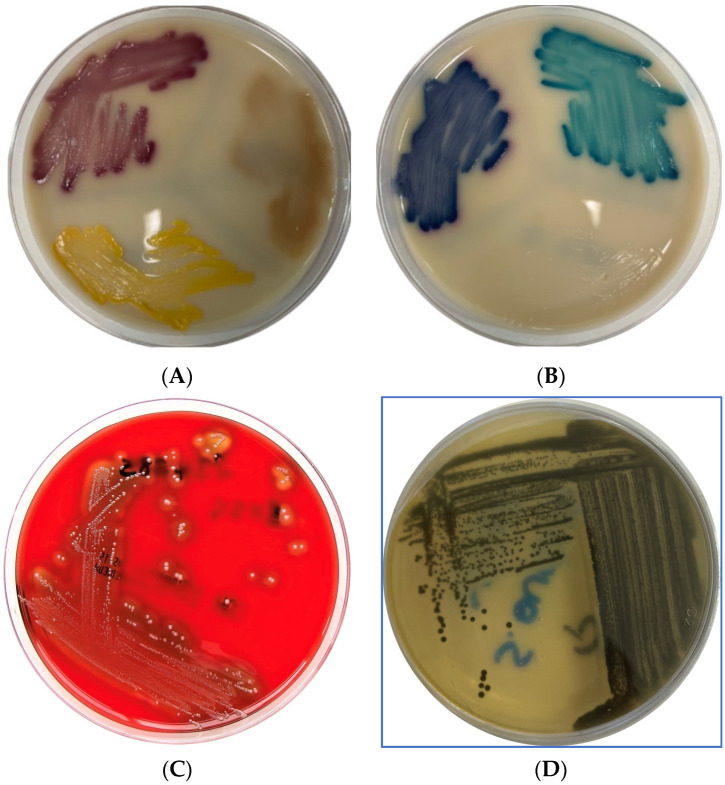
Typical culture characteristics and colors of the bacteria analyzed in this study. (**A**) UTI medium *E. coli*—pink/purple, *P. aeruginosa*—brown translucent, and *S. aureus*—yellow; (**B**) *E. faecalis*—turquoise, *K. pneumoniae*—dark blue; (**C**) haemolytic *Streptococcus* spp. on Columbia CAN with 5% sheep blood and (**D**) *S. aureus* on Baird-Parker medium.

**Table 1 ijms-25-03121-t001:** Genera of bacteria isolated from wounds of companion animals (n = 136).

**Gram-negative (n = 65; 47.79%)**		
**Genus**	**Number**	**Percentage**
*Acinetobacter*	10	7.35
*Aeromonas*	1	0.74
*Brevundimonas*	2	1.47
*Citrobacter*	3	2.21
*Enterobacter*	3	2.21
*Escherichia*	11	8.09
*Hafnia*	1	0.74
*Klebsiella*	3	2.21
*Leclercia*	1	0.74
*Pantoea*	3	2.21
*Moraxella*	1	0.74
*Proteus*	8	5.88
*Pseudomonas*	9	6.62
*Psychrobacter*	3	2.21
*Serratia*	4	2.94
*Stenotrophomonas*	2	1.47
**Gram-positive (n = 71; 52.21%)**		
*Bacillus*	1	0.74
*Curtobacterium*	1	0.74
*Enterococcus*	17	12.50
*Kocuria*	1	0.74
*Lactococcus*	1	0.74
*Lysinibacillus*	1	0.74
*Macrococcus*	1	0.74
*Micrococcus*	1	0.74
*Microbacterium*	4	2.94
*Peribacillus*	1	0.74
*Staphylococcus*	37	27.21
*Streptococcus*	5	3.68

**Table 2 ijms-25-03121-t002:** The resistance profile in the dominant groups of bacteria isolated from wounds of companion animals; 0–8: number of antibiotics bacteria in different groups were resistant to *.

	Number of Antibiotics Bacteria Are Resistant to (n, %)									
Group of Bacteria	0	1	2	3	4	5	6	7	8	MDR
*Enterobacterales* (n = 38)	3 (7.9)	2 (5.3)	12 (31.6)	7 (18.4)	6 (15.8)	3 (7.9)	2 (5.3)	2 (5.3)	1 (2.6)	16 (42.1)
*Pseudomonas* (n = 7)	0	0	0	1	0	3 (42.9)	3 (42.9)	0	0	7 (100)
*Acinetobacter* (n = 9)	0	5 (55.6)	1 (11.1)	1 (11.1)	1 (11.1)	1 (11.1)	0	-	-	3 (33.3)
*Enterococcus* (n = 15)	0	3 (20.0)	3 (20.0)	2 (13.3)	7 (46.7)	0	-	-	-	9 (60)
*Staphylococcus* (n = 51)	14 (27.5)	15 (29.4)	8 (15.7)	3 (5.9)	5 (9.8)	0	3 (5.9)	3 (5.9)	0	14 (27.5)

* Values in the table show the number and percentage (in brackets) of bacteria resistant to different numbers of antimicrobial agents. For example, in *Enterobacterales*, there were three (7.9%) bacteria susceptible to all antibiotics tested. Two (5.3%) bacteria were resistant to one antibiotic, 12 (31.6%) bacteria were resistant to two antimicrobials, etc. One isolate of *Enterobacterales* (comprising 2.6%) was resistant to a total of eight antibiotics.

**Table 3 ijms-25-03121-t003:** Positive results for Gram-positive (n = 59) bacteria isolated from cats (n = 25) and dogs (n = 34).

Animal	n (%)					
	*mecA*	*msrA*	*lnuA*	*strA*	*tetK*	*sul3*
Cat	0	3 (12)	0	7 (28)	6 (21.4)	6 (21.4)
Dog	4 (11.8)	1 (2.9)	4 (11.8)	10 (29.4)	3 (8.8)	3 (8.8)

**Table 4 ijms-25-03121-t004:** Positive results for Gram-negative (n = 49) bacteria isolated from cats (n = 12), dogs (n = 33), and rabbits (n = 4).

Animal	n (%)						
	*blaTEM*	*blaSHV*	*blaCTX-M*	*blaOXA-1*	*sul3*	*qnrD*	*strA*
Cat	3 (25)	0	1 (8.3)	0	3 (25)	0	1 (8.3)
Dog	10 (30.3)	3 (9.1)	1 (3)	1 (3)	2 (6.1)	0	10 (30.3)
Rabbit	0	1 (25)	0	0	0	0	1 (25)

**Table 5 ijms-25-03121-t005:** Selected phenotypic and molecular resistance profiles of canine and feline bacteria isolated from wounds.

Origin	Species	Phenotype of Resistance (Antibiotic Class)	Resistance Genes (Type of Resistance)
feline	*Enterococcus faecalis*	IMP (β-lactam-carbapenem) ENR (fluoroquinolone) TGC (tetracycline) TY (macrolide)	*mecA* (methicillin) *msrA* (macrolides) *lnuA* (lincosamides) *tetK* (tetracyclines)
canine	*Enterococcus faecalis*	IMP (β-lactam-carbapenem) ENR (fluoroquinolone) TGC (tetracycline) TY (macrolide)	*msrA* (macrolides) *strA* (aminoglycosides) *tetK* (tetracyclines) *sul3* (sulfonamides)
canine	*Staphylococcus sciuri*	-	*lnuA* (lincosamides) *strA* (aminoglycosides) *tetK* (tetracyclines)
canine	*Staphylococus pseudintermedius*	TE (tetracycline) DA (lincosamide) E (macrolide) SXT (diaminopyrimidines/sulfonamide) ENR (fluoroquinolone) CN (aminoglycoside) TY (macrolide)	*lnuA* (lincosamides) *strA* (aminoglycosides) *tetK* (tetracyclines)
canine	*Escherichia coli*	CN (aminoglycoside) AMC (β-lactam/β-lactamase inhibitor) AMP (β-lactam-aminopenicillin) IMP (β-lactam-carbapenem)	*blaTEM* (ESBL) *blaSHV* (ESBL) *strA* (aminoglycosides)
canine	*Escherichia coli*	CTX (β-lactam–3rd gen. cephalosporin) AMC (β-lactam/β-lactamase inhibitor) CAZ (β-lactam–3rd gen. cephalosporin) AMP (β-lactam-aminopenicillin) TY (macrolide)	*blaTEM* (ESBL) *blaSHV* (ESBL) *strA* (aminoglycosides)
canine	*Proteus mirabilis*	CN (aminoglycoside) SXT (diaminopyrimidines/sulfonamide) CTX (β-lactam–3rd gen. cephalosporin) AMC (β-lactam/β-lactamase inhibitor) CAZ (β-lactam–3rd gen. cephalosporin) ENR (fluoroquinolone) AMP (β-lactam-aminopenicillin) IMP (β-lactam/carbapenem)	*blaTEM* (ESBL) *blaOXA-1* (ESBL-carbapenems) *strA* (aminoglycosides)

**Table 6 ijms-25-03121-t006:** Antimicrobial susceptibility tests according to groups of bacteria.

	*Enterobacterales* (*E. coli*, *Klebsiella*, *Proteus*, *Enterobacter*)	*Pseudomonas*	*Acinetobacter*	*Enterococcus*	*Staphylococcus*
No of strains in total	38	7	9	15	51
Cats	7	1	3	5	21
Dogs	28	6	5	10	30
Rabbit	3	0	1	0	0
antimicrobial disks abbreviations *	ENR	ENR	ENR	ENR	ENR
	AMC (ESBL) **	AMC (ESBL) **	AK	AMP	E (MLSb) **
	CAZ (ESBL) **	CAZ (ESBL) **	CN	MEM/IMP	DA (MLSb) **
	CTX (ESBL) **	CTX (ESBL) **	MEM/IMP	TGC	FOX (MRS) **
	AMP	AK	SXT	TY	CN
	CN	MEM/IMP	TY		SXT
	SXT	TZP			TE
	MEM/IMP	TY			TY
	TY				

* ENR (enrofloxacin 5 µg), E (erythromycin 15 µg), DA (clindamycin 2 µg), FOX (cefoxitin 30 µg), CN (gentamicin 10 µg), SXT (trimethoprim/sulfamethoxazole 1.25/23.75 µg), TE (tetracycline 30 µg), TY (tylosin 30 µg), AMC (amoxicillin/clavulanic acid 20/10 µg), CAZ (ceftazidime 30 µg), CTX (cefotaxime 30 µg), AMP (ampicillin 10 µg), MEM (meropenem 10 µg), IMP (imipenem 10 µg), TZP (piperacillin/tazobactam 100/10 µg), TGC (tigecycline 15 µg). ** in brackets—mechanisms of resistance determined using the provided antimicrobial discs: ESBL—extended spectrum beta lactamases; MLSb—constitutive or inducible resistance to macrolides, lincosamids, and streptogramins b; MRS—methicillin resistance.

**Table 7 ijms-25-03121-t007:** PCR primers used in the study.

No.	Gene	Primer	Sequence (5′-3′)	Annealing Temp. (°C)	Product Length (bp)	Reference
1.	*msrA*	msrA-F	GGCACAATAAGAGTGTTTAAAGG AAGTTATATCATGAATAGATTGTCCTGTT	50	940	[19]
		msrA-R				
2.	*ereA*	ereA-F	AACACCCTGAACCCAAGGGACG CTTCACATCCGGATTCGCTCGA	57	420	[20]
		ereA-R				
3.	*lnuA*	lnuA-F	GGTGGCTGGGGGGTAGATGTATTAACTGG GCTTCTTTTGAAATACATGGTATTTTTCGATC	57	323	[19]
		lnuA-R				
4.	*mecA*	mecA-F	GTAGAAAATGACTGAACGTCCGATAA CAATTCCACATTGTTTCGGTCTAA	55	310	[21]
		mecA-R				
5.	*tetK*	tetK-F	TCGATAGGAACAGCAGTA CAGCAGATCCTACTCCTT	55	169	[22]
		tetK-R				
6.	*blaTEM*	blaTEM-F	ATTCTTGAAGACGAAAGGGC ACGCTCAGTGGAACGAAAAC	60	1150	[23]
		blaTEM-R				
7.	*blaSHV*	blaSHV-F	CACTCAAGGATGTATTGTG TTAGCGTTGCCAGTGCTCG	52	885	[23]
		blaSHV-R				
8.	*blaCTX-M*	blaCTX-M-F	CGATGTGCAGTACCAGTAA TTAGTGACCAGAATCAGCGG	55	585	[24]
		blaCTX-M-R				
9.	*blaOXA-1*	blaOXA-1-F	ACACAATACATATCAACTTCGC AGTGTGTTTAGAATGGTGATC	61	813	[23]
		blaOXA-1-R				
10.	*sul3*	sul3-F	ACCACCGATAGTTTTTCCGA TGCCTTTTTCTTTTAAAGCC	62	199	[25]
		sul3-R				
11.	*qnrA*	qnrA-F	GGGTATGGATATTATTGATAAAG CTAATCCGGCAGCACTATTA	55	580	[26]
		qnrA-R				
12.	*qnrD*	qnrD-F	AGTGAGTGTTTAGCTCAAGGAG CAGTGCCATTCCAGCGATT	53	175	[27]
		qnrD-R				
13.	*strA*	strA-F	TCAATCCCGACTTCTTACCG CACCATGGCAAACAACCATA	52	126	[28]
		strA-R				

## Data Availability

The data generated in this study are available on request from the corresponding author.

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
