# Peer review of "Wounds of Companion Animals as a Habitat of Antibiotic-Resistant Bacteria That Are Potentially Harmful to Humans—Phenotypic, Proteomic and Molecular Detection"

_ijms, 2024, doi:10.3390/ijms25063121_

Round 1
Reviewer 1 Report
Comments and Suggestions for Authors
This article deals with the study of bacterial infections in pet animals wounds and their susceptibillity to the common and last resort antibiotics and the relative presence of resistance genes, in order to establish, even for the animals, a correct and appropriate therapy able to combact pet infections that could be transmitted to their owners and to the surrounding environment, quickly spreading the infection. In fact using for the animals an empiric treatment as often happens, may be harmful leading to selection of resistant bacteria that would be able to infect human population. From this point of view, this paper turns out to be quite interesting. However the authors only considered the microorganisms isolated from animals without no reference to humans. I mean that in no way the research has been conducted including a human population and determining the correspondence with the relative animals infection or colonization . I suppose that perhaps the authors in this article have only considered the animals infection and later in a subsequent study they would possibly compare the bacterial flora present in wounds infection in animals with men and children living together with them. I would like the authors to specify this concept. Below I list the problems found in the article: -the layout is incorrect, in fact the number of pages is not listed consecutively and lines are reported up to 125. - for some acronyms the explanation is not reported i.e. ARD in line 32 or MRS in table2 (this is reported only subsequently in the legenda of table 6 and not at the first mention) - there are some inconsistencies, so in lines 84-86 it is reported "None of the examined antimicrobials were effective against all strains" and immediately after "only amikacin proved effective against all tested strains" . Please clarify -the number of examined samples were not clearly specified. Are they 136, 120 or even 145? -is the reported concept that the selection pressure of bacteria in farm poultry is higher than bacteria derived from pet animals, correct and what is the reason? -the term Enterobacteriaceae in the section "Materials and Methods" should be substituted with the term Enterobacterales
.
Comments on the Quality of English LanguageThere are some minor errors. The correct name of antibiotics is imipenem and meropenem and not imipeneme and meropeneme. Some sentences should be revised: for example the heading of table 2 (it could be written in a clearer way) or in the second page of the section Discussion (second and third line) and in the second paragraph of the Discussion ("and their owners. [13] examined the presence...".)
Author Response
We would like to thank the Reviewer for the helpful suggestions. Below are the point-by-point answers to the Reviewer’s remarks;
- However the authors only considered the microorganisms isolated from animals without no reference to humans. I mean that in no way the research has been conducted including a human population and determining the correspondence with the relative animals infection or colonization . I suppose that perhaps the authors in this article have only considered the animals infection and later in a subsequent study they would possibly compare the bacterial flora present in wounds infection in animals with men and children living together with them. I would like the authors to specify this concept.
Reply: Our major goal was to determine the composition of bacteria infecting wounds of companion animals, not the common carriage (as specifically stated in the Abstract, key words and the Introduction). In such case, the most commonly found microorganisms are either animal or human/common pathogens, not the commensal species. Actually, the presented study was indeed an introductory research, but the latter research was focused on the development of preparations that could be used in prevention of wound infections. Of course, the suggested examination (i.e. the potential of the isolated and identified strains to infect humans) would be an interesting supplement to these presented in the manuscript. However, while trying to search for the common microorganisms, it would be necessary to isolate bacteria from various parts of the human body, still with quite high possibility of not reaching the site where the microorganism would be present. Hence, the risk of false negative results and distorted analytical results. It would be a completely different matter if the carrier status would have been tested – then swab samples would be taken only from e.g. nose of people/pets. What also needs to be remembered in that place is that the carriage of microorganisms (e.g. S. aureus, group B. streptococci) is temporary in humans, so this is another factor of possible false negative results. With the fact that wounds were located very differently, there are very few tangent points. We removed some fragments of the text that would suggest that we aim at describing the pets’ owners testing in our next manuscript (e.g. Introduction, line 71-74). .
- -the layout is incorrect, in fact the number of pages is not listed consecutively and lines are reported up to 125
Reply: This problem occurred during formatting of the text (i.e. tables in horizontal layout, Materials and Methods at the end of the text, instead of before Results). Of course, this has been corrected.
- for some acronyms the explanation is not reported i.e. ARD in line 32 or MRS in table2 (this is reported only subsequently in the legenda of table 6 and not at the first mention)
Reply: We added the missing explanations of acronyms at their first mention in the text.
- there are some inconsistencies, so in lines 84-86 it is reported "None of the examined antimicrobials were effective against all strains" and immediately after "only amikacin proved effective against all tested strains"
Reply: The mentioned fragment was supposed to refer to the effect of amikacin against Pseudomonas spp. strains (where indeed all strains of Pseudomonas proved susceptible to this antibiotic). We corrected this sentence.
- Please clarify -the number of examined samples were not clearly specified. Are they 136, 120 or even 145?
Reply: The number 145 is the total number of collected swabs from animals (as stated in the “Collection of samples” section. The total number of isolated bacterial strains was finally 136. All these bacteria were subjected to the MALDI-TOF tests in order to identify their species. Having regard to the fact that some species (e.g. Brevundimonas diminuta, Hafnia alvei, Bacillus pumilus or Curtobacterium flaccumfaciens) proved to be either commensal bacteria, or typically environmental, there are no EUCAST or other international recommendations for antimicrobial susceptibility testing available for these bacteria. With also the fact that some wounds were abundant in bacterial species, we assumed that these are not the causal agents of infection. For this reason, they were excluded from further analysis. We added this explanation in the Materials and Methods section (4.3. Antibacterial susceptibility tests).
- is the reported concept that the selection pressure of bacteria in farm poultry is higher than bacteria derived from pet animals, correct and what is the reason?
Reply: We provided the reasons for the selection pressure on bacteria in farm poultry being higher than in pet animals (i.e. the doses of antimicrobial agents administered to poultry are often inflated, used contrary to the veterinarians’ instructions or the treatment duration is extended without justification) and provided the reference.
- -the term Enterobacteriaceae in the section "Materials and Methods" should be substituted with the term Enterobacterales
Reply: Corrected
- There are some minor errors. The correct name of antibiotics is imipenem and meropenem and not imipeneme and meropeneme.
Reply: Corrected
- Some sentences should be revised: for example the heading of table 2 (it could be written in a clearer way) or in the second page of the section Discussion (second and third line) and in the second paragraph of the Discussion ("and their owners. [13] examined the presence...".)
Reply: The mentioned sentences were corrected as well as other sentences that needed correction (the manuscript was double-checked for the sentences that needed clarification).
We hope that our replies and the corrected version of the manuscript will be found acceptable by the Reviewer.
Reviewer 2 Report
Comments and Suggestions for Authors
This manuscript has to be improved a lot. There are many obvious mistakes, for example:
Line 32, ARD should be ARG;
Line 79-96: All figure numbers seem wrong, the explanation of figures didn't correspond to the right figures in the manuscript.
There are no line numbers from the ninth page, and the figure numbers are also wrong in the following text such as in the section 4.
Besides the above mistakes, the quantitative data is lack, for example, there was no qPCR data of ARGs. Moreover, qualities of figures should be improved a lot.
A total of 136 bacterial strains were isolated from wound swabs, and the strains were identified and tested their resistance to antibiotics. Moreover, 12 ARGs were detected using PCRs.This manuscript is meaningful for understanding the risk of ARB and ARG transfered to the owners of pets. However, as I have mentioned before, there are many mistakes in this manuscirpt, and the figures is not clear and the quality should be improved. Furthermore, the number of isolated strains is 136 firstly, and then it changed to 120 for the test of antimicrobial resistance. The reason is not mentioned. Especially, the order of figure numbers in the text all seem wrong, which affect my understanding of this manuscript. Comments on the Quality of English Language
Moderate editing of English language required.
Author Response
We would like to thank the Reviewer for the helpful suggestions. Below are the point-by-point answers to the Reviewer’s remarks;
- Line 32, ARD should be ARG;
Reply: Corrected
- Line 79-96: All figure numbers seem wrong, the explanation of figures didn't correspond to the right figures in the manuscript.
AND
- There are no line numbers from the ninth page, and the figure numbers are also wrong in the following text such as in the section 4.
Reply: It appeared that during providing the tables in pages the layout of which needed to be horizontal, we divided the manuscript into sections and during this division a lot of formatting went wrong (i.e. page numbers, line numbering). Then, while formatting the manuscript for the layout typical for IJMS, the Materials and Methods section was transferred to the end of the paper (before conclusions), and I mistakenly corrected only the figure numbers, without correcting them in the text. Of course, all these mistakes are corrected now: the page numbering is correct, the line numbers are continuous from the first to the last page and the figure as well as table numbers agree with their reference in the text.
- Besides the above mistakes, the quantitative data is lack, for example, there was no qPCR data of ARGs.
Reply: Firstly, we cannot agree with the above remark. Firstly, there are a lot of quantitative data, such as:
- a) the number of strains identified as different bacterial species;
- b) the percentage of isolates from various species in which the resistance(including different mechanisms of resistance such as ESBL, MRS, MSb, MLSb) to the examined antibiotics was observed;
- c) number and percentage of bacterial strains in which the examined genetic determinants were detected.
Secondly, in our experimental setting no information will be gained by the use of qPCR. Even if an assay was designed and calibrated to reliably measure (i.e. quantify) the amount of physical copies of antimicrobial resistance genes, in various often phylogenetically remote species of bacteria, it would only tell us how many copies of a given gene was present in a portion of medium-grown bacterial colony used for the DNA extraction. This information has no biological significance, as the relevant would be the amount of bacteria in a source wound, which could be much more easily estimated using culture-based methods. If a pure colony was isolated from a wound and it carries an ARG, then it means that the source bacterium also carries this gene, which may have its consequences in the form of the ARG transfer. And this particular trait of bacteria has its consequences, this obtaining it is scientifically sound.
- Moreover, qualities of figures should be improved a lot.
Reply: We increased the quality of figures and supplied them as separate files (perhaps this will be the best solution to have the best possible quality of figures). The figures 1, 4 and 6 (in the initial version) were prepared in a collage creator, so probably this caused the decrease in figure quality. In the revised version the figures will be pasted into the main text without the collage (for the Reviewers and Editor to have the better view of the corrected figures, with correct numbers) and apart from that – the figures will be provided as separate files in the submission system.
- Furthermore, the number of isolated strains is 136 firstly, and then it changed to 120 for the test of antimicrobial resistance. The reason is not mentioned.
Reply: The total number of bacterial strains isolated from swab samples was 136. All these bacteria were subjected to the MALDI-TOF tests in order to identify their species. Having regard to the fact that some species (e.g. Brevundimonas diminuta, Hafnia alvei, Bacillus pumilus or Curtobacterium flaccumfaciens) proved to be either commensal bacteria, or typically environmental, there are no EUCAST or other international recommendations for antimicrobial susceptibility testing available for these bacteria. With also the fact that some wounds were abundant in bacterial species, we assumed that these are not the causal agents of infection. For this reason, they were excluded from further analysis. We added this explanation in the Materials and Methods section (4.3. Antibacterial susceptibility tests).
- Especially, the order of figure numbers in the text all seem wrong, which affect my understanding of this manuscript.
Reply: As clarified before (response to remark No. 2 and 3), while formatting the manuscript for the layout typical for IJMS, the Materials and Methods section was transferred to the end of the paper (before conclusions), and I mistakenly corrected only the figure numbers, without correcting them in the text. Of course, all these mistakes are corrected now: the page numbering is correct, the line numbers are continuous from the first to the last page and the figure as well as table numbers agree with their reference in the text.
- Moderate editing of English language required.
Reply: The text has been double checked and the language issues, or sentences that needed rephrasing were corrected.
We hope that our replies and the corrected version of the manuscript will be found acceptable by the Reviewer.
Reviewer 3 Report
Comments and Suggestions for Authors
The paper is written correctly, however standard methods in veterinary clinical bacteriology were used in the research. The work lacks the sequencing of a certain number of isolates, in order to carry out the genetic characterization of a certain number of strains and to define certain genes. For example, blaTEM gene was detected, but which one, there are many groups and subtypes of blaTEM genes, etc. I think that the publication of only the resistotype with the confirmation of the resistance gene by the PCR method is insufficient.
Abstract
* The abstract is written correctly, clearly and concisely with relevant facts obtained from the this research
Introduction
* In the introduction, the authors presented the current situation and problems faced by veterinarians in the antibiotic therapy of skin infections in dogs and cats caused by different strains of bacteria. Special emphasis was placed on the transmission of strains with resistance genes between pet owners and dogs and cats.
Materials and Methods
The samples were collected with a sterile swab by a qualified veterinarian (Fig. 1)...
* Fig 1 does not show sampling with steril swab but Fig 5 does.
Figure 6. Typical colonies of (A) E. coli, P. aeruginosa and S. aureus, (B) E. faecalis, K. pneumoniae, (C) Staphylococcus sp. (C) and S. aureus (D), grown on UTI medium (A and B) and on Baird-Parker medium (C and D).
* The photos are not representative for the publication, they are blurry and the bacterial strains are seeded very densely, individual cultures are not visible, in order to obtain clear morphological characteristics of the colonies on the media. Also, the photos look like they were taken from the internet. The suggestion to the authors is to make better photos or to remove them from the publication.
Figure 7. Light microscope image (1000 ×) of Pseudomonas aeruginosa (on the left) and Staphylococcus aureus (on the right).
* Morphology of Pseudomonas aeruginosa and Staphylococcus aureus on the light microscope image is known and I suggest to the authors that it be removed from the publication. It does not need to be in a publication.
Results
A total of 136 bacterial isolates were obtained from the collected swabs, including: 52 (38.24%) from cats, 80 (58.82%) from dogs and 4 (2.94%) from rabbits.
* It should be shown from the largest number to the smallest, first the results for the dogs, then the cats and finally the rabbit.
Table 1. Species of bacteria isolated from wounds of three groups of animals.
* Table 1 is very large, confusing and unreadable for publication... it is better to include the results from the table in text, because this kind of display disturbs the reader, it is difficult to follow... avoid such large tables.
You isolated 136 isolates in total, and you tested the antimicrobial susceptibility of 120 isolates....
* What happened to the 16 other isolates, were they sensitive??? It should be presented in one sentence, why you selected 120 isolates for further research.
Have you only used imipenem?
* Why didn't you use meropenem?
Figure 1. The resistance mechanisms observed in the examined groups of bacteria. A – Extended Spectrum Beta Lactamases (ESBL) in Enterobacterales; B – methicillin resistance in Staphylococcus pasteuri; C – inducible resistance to macrolides, lincosamids and streptogramins b (MLSb) in Staphylococcus pseudintermedius; D – constitutive MLSb in S. pseudintermedius.
* Why do you put a large number of antibiotic discs on one plate. Up to five discs is enough... Is it so described in your standard operating procedure???
Discussion
* Discussion is correct...
* Do clinical veterinarians in Poland comply with Regulation (EU) 2019/6 on Veterinary Medicinal Products which provides a wide range of concrete measures for the rational use of antibiotics in veterinary medicine (category A "avoid antibiotics", category B "restricted-use antibiotics", category C "caution antibiotics" and category D "prudence antibiotics").
Author Response
Dear Editor,
Thank you for your comments. Please find the replies in the attached Word document.
We hope that you find the replies acceptable.
With kind regards,
Anna Lenart-Boroń

Round 2
Reviewer 2 Report
Comments and Suggestions for Authors
Authors have addressed all the questions, and the quality of the manuscript has been improved. However, there are still some issues in the data presenting as following:
(1) Figure 1D: The figure caption isn’t corresponded to the words in pictures. Moreover, this figure couldn't stand for resistance mechanisms, it's only show the results of antibacterial susceptibility tests. It's suggested to be put in the supplementary information (SI).
(2) Are the data in Table 2 same with Figure 2 and 3? And what's the meaning of R0 to R8? If they are the same data, there is no need to show data in the table.
(3) Line 179, ten out of the tested 12 genetic determinants of antimicrobial resistance were detected, why figure 4 only showed the results of two genes? Furthermore, what's the meaning of number 1-18 and 1-24 in figure 4? This picture is not explained clearly.
(4)Figure 5 was a photo of sampling, which was also suggested to be in the SI.
(5) The names of bacterial strains in Figure 6 are not clear.
Comments on the Quality of English LanguageEnglish has been improved.
Author Response
We would like to thank the Reviewer for the further suggestions. Below are the point-by-point answers to the Reviewer’s remarks:
(1) Figure 1D: The figure caption isn’t corresponded to the words in pictures. Moreover, this figure couldn't stand for resistance mechanisms, it's only show the results of antibacterial susceptibility tests. It's suggested to be put in the supplementary information (SI).
We have changed the figure caption, both in terms of the bacterial names and we changed “mechanisms” to phenotypes. We also transferred this figure to Supplementary Information.
(2) Are the data in Table 2 same with Figure 2 and 3? And what's the meaning of R0 to R8? If they are the same data, there is no need to show data in the table.
Indeed, data such as ESBL, MRS, MLS were the same as in Figures, so we deleted them. We added explanation about the meaning of R0-R8 and in the Table footnote we added a brief explanation of the exact meaning of values presented in Table 1.
(3) Line 179, ten out of the tested 12 genetic determinants of antimicrobial resistance were detected, why figure 4 only showed the results of two genes? Furthermore, what's the meaning of number 1-18 and 1-24 in figure 4? This picture is not explained clearly.
Indeed, the citation of Figure 4 suggests that it should present all ten out of 12 genes. It of course presents the selected genes. We corrected this, by moving the citation of this figure to lines where the specific genes are mentioned; we replaced lnuA (which was not the most frequently detected gene) with strA and sul, which were the most frequently observed ones. Now, after correction, this figure presents the three most frequently detected genes in this study and the three the frequent presence of which was described in the text. The description of numbers 1-18 and 1-24 was already given in the previous version. These are gel lanes, where DNA extracts of individual bacteria were shown.
(4)Figure 5 was a photo of sampling, which was also suggested to be in the SI.
The Figure 5 was transferred to Supplementary Data (Supplementary Figure 2).
(5) The names of bacterial strains in Figure 6 are not clear.
We have provided photos for Picture 6A and 6B without names of bacteria and described them in the figure caption. We have also corrected the caption for Figure 6 (now Figure 4 after two figures were removed from the manuscript text).
Comments on the Quality of English Language
English has been improved.
Response: Thank you
We hope that our replies and the corrected version of the manuscript will be found acceptable by the Reviewer.
Round 3
Reviewer 2 Report
Comments and Suggestions for Authors
Authors have improved tha manuscript according previous suggestions. Now there are still two small questions about the response as following:
(1) Table 2 : if the R0-R8 mean the number of antibiotics, only the number of 0-8 used is more easier to understand.
(2) The quality of Figure 3 of positive results of PCR test should be improved, and why they have different number of lanes? Furthermore, PCR test results of other genes should be added in the supplementary information.
